# Effect of Ospemifene on Densitometric and Plasma Bone Metabolism Biomarkers in Postmenopausal Women Reporting Vulvar and Vaginal Atrophy (VVA)

**DOI:** 10.3390/jcm11216316

**Published:** 2022-10-26

**Authors:** Silvia Maffei, Letizia Guiducci

**Affiliations:** 1Department of Cardiovascular Endocrinology and Metabolism, Gynaecological and Cardiovascular Endocrinology and Osteoporosis Unit, “Gabriele Monasterio” Foundation and Italian National Research Council (CNR) Pisa, 56124 Pisa, Italy; 2National Research Council, Institute of Clinical Physiology, Via Moruzzi 1, 56124 Pisa, Italy

**Keywords:** bone metabolism, postmenopausal women, ospemifene

## Abstract

Menopausal hormone deficiency can exert multiple effects on various organs. Vulvovaginal atrophy (VVA) is among the most widespread and disabling post-menopausal disorder. Hormonal changes can also result in a markedly increased rate of bone mineral density (BMD) loss. Ospemifene (OSP) is an SERM indicated to treat vulvar and vaginal atrophy (VVA) in postmenopausal women. This study evaluates the long-term effects of ospemifene therapy on bone metabolism and bone mineral parameters in postmenopausal women reporting VVA/GSM. Methods: Women reporting VVA symptoms were included. Bone health profile was investigated in 61 subjects treated with OSP (OSPG) (60 mg/day) and compared with a control group (CG) (*n* = 67) over 12 months. Results: In the CG, BMD and T-score statistically decreased at the femoral neck (FN), total femur (TF), and lumbar spine (L1–L4). In the OSPG, BMD decreased significantly at FN but tended to remain stable at TF and L1–L4. No changes were observed in bone mineral markers after one year in either group, except BAP, which decreased in OSPG. Conclusions: Long-term OSP treatment improves bone mineral markers at TF and LS and slows bone loss at FN compared to the control group. Overall, OSP exerts a protective effect on bone loss in healthy menopausal women with VVA.

## 1. Introduction

Alteration of the hormone profile during and after the onset of menopause can exert multiple effects on various organs. Low estrogen levels can lead to atrophy of hormone-dependent tissue areas (vagina and vulva), inducing vulvo-vaginal syndrome (VVA) or genital-urinary syndrome (GSM).

Vulvovaginal atrophy (VVA) is one of the most widespread and disabling post-menopausal disorders [1].

Vaginal dryness and dyspareunia are the most common symptoms of VVA reported by post-menopausal women [2].

Menopausal hormonal changes can also result in a markedly increased rate of bone mineral density (BMD) loss, particularly during the three years surrounding the final menstrual period [3]. The decline of the estrogen milieu induces increased osteoclast activity and bone resorption, which can lead to reduced skeletal mass and osteopenia or osteoporosis.

Osteoporosis is a progressive systemic skeletal disease characterized by low bone mass and microarchitectural deterioration of bone tissue [4]. Postmenopausal osteoporosis is of particular concern because it leads to an increased risk of fractures, with subsequent negative impacts on health in older women. 

Therefore, menopause could be considered a time-limited window of opportunity to treat and/or prevent rapid bone loss and trabecular microarchitectural damage and, thus, prevent osteoporosis in later years. [5]. Many studies confirm the efficacy of different hormone replacement therapy (HRT) regimens for the relief of climacteric syndrome in the treatment of osteoporosis [6]. Osteoporosis prevention can actually be considered as an important additional effect in women who use HRT for the treatment of climacteric symptoms. Bone protection is one of the major benefits of HRT. Nevertheless, not all menopausal women are eligible for HRT; some women refuse HRT for fear of breast cancer; for some, HRT is contraindicated. There is, therefore, a need for new therapeutic options for these women during menopause.

The selective estrogen receptor modulator (SERM) ospemifene (OSP) has recently become available as a new therapeutic option for the treatment of VVA in postmenopausal women [7,8]. Indeed, OSP is specifically indicated for the treatment of VVA due to its particular stimulating effect on the hormone receptors of the vaginal mucosa, but has a neutral effect on the estrogen receptor in the uterus and antagonist effect on the estrogen receptor in the breast. [9,10]. Ospemifene is also currently the only drug with FDA and EMA approval for the treatment of female breast cancer survivors after the treatment of breast cancer, including adjuvant therapy, has been completed [11]. The safety profile of ospemifene treatment on clotting and thrombotic markers and on metabolic profile has been established in numerous studies [12,13]. 

Few studies focus on bone biochemical markers and bone quality, and data collection is limited to three or six months [14,15,16]. The possibility that ospemifene causes a decrease in fracture risk is not yet proven, but the scientific evidence is compelling. 

The aim of the present study was to evaluate the long-term effects of ospemifene therapy on bone metabolism and bone mineral parameters in postmenopausal women reporting VVA/GSM. 

## 2. Materials and Methods

### 2.1. Study Design 

The study was conducted at the Endocrinological–Cardiovascular Gynaecology Outpatient Clinic and the Osteoporosis Study Centre of the Italian National Research Council Gabriele Monasterio Tuscan Foundation in Pisa, Italy. The study was authorized by the local ethics committee (Prot No. 37981, study 3605, 20 June 2012). The study was registered at ClinicalTrials.gov with the identification code NCT03699150. 

Women who reported symptoms related to VVA were included in a study to evaluate the effects of ospemifene on cardiometabolic risk [12]. Women with current cancer, thromboembolic disease, current or previous liver disease, who had used steroids in the previous three months, or who had received hormonal therapies in the six months prior to the start of the study were excluded. In addition, women with a BMI equal to or greater than 30 kg/m^2^ were excluded, since there are no data in the literature relating to the effects of ospemifene in obese patients or with risk factors for thromboembolism. Of the women participating in this wider study for at least 12 months, the bone health profile was investigated in a subgroup of 61 subjects (OSPG) and compared with a control group (CG) (*n* = 67) over 12 months of treatment.

In this interventional and prospective-control study, ospemifene was administered continuously at a daily dose of 60 mg orally to postmenopausal women with VVA, and bone parameters were compared before and after treatment with a matched control group. In both groups, studied bone parameters were evaluated at recruitment (T0), and 12 months (T1) of treatment. 

When necessary, treatment with vitamin D supplementation was carried out in patients of both groups to restore levels to adequate values (>30 ng/mL). The dosage of vitamin D supplements varied according to baseline values. 

Clinical and laboratory assessments were as follows. Bone mineral density (BMD) was assessed at L1–L4, FN, and TF through dual-energy X-ray absorptiometry (DEXA, Explorer QDR Series bone densitometer, Hologic, Marlborough, MA, USA) according to the manufacturer’s instructions, which included a quality control test using a standard phantom. The DEXA was performed at T0 and at T1. The T-score represents the ratio between the bone mineral content and the area in square centimeters and is expressed as a standard deviation score from a normal reference population database [17]. Data were classified as follows: T − score ≥ −1 = normal, −1 > T − score > − 2 5 = low bone density (osteopenia), and T − score ≤ − 2 5 = OP [17]. The Z-score represents the ratio between the bone mineral content and the area in square centimeters and is expressed as a standard deviation score between the patient and normal controls of the same age and gender [17].

Fasting blood sampling for the evaluation of bone metabolism biomarkers and medical examinations were performed between 7.00 and 9.30 am at T0 and T1. 

Specifically, blood samples were taken after an overnight fast and centrifuged at 2500× *g* for 10 min. Samples were then assayed for the following biomarkers: total calcium (t-Ca; heparinized plasma; biochemistry, using a CX9 Chemistry Analyzer, Beckman, CA, USA), ionized calcium (i-Ca; external reference laboratory), serum vitamin D (25(OH)D; Liason, DiaSorin, Italy), bone alkaline phosphatase (BAP; Liason, DiaSorin, Italy), osteocalcin (OC; Liason, DiaSorin, Italy), parathyroid hormone 1-84 (PTH; sample maintenance at 4 °C, plasma EDTA, Liason, DiaSorin, Italy), and phosphorus (sample maintenance at 4 °C, serum, Architect, Abbott).

Data on family history and comorbidities were also recorded, namely hypertension, dyslipidemia and hypercholesterolemia, type 2 diabetes, and hypothyroidism 

### 2.2. Patients 

Mean age at the start of study was 54.8 ± 5.0 years in the OSPG and 57.6 ± 5.3 years in the CG (*p* = NS). The BMI was 22.8 ± 3.5 kg/m^2^ in the OSPG and 23.5 ± 2.5 kg/m^2^ in the CG (*p* = NS). The presence of a family history of osteoporosis was 36.4% in the OSPG and 33.7% in the GG. 

### 2.3. Statistical Analysis 

Continuous variables were reported as the mean ± standard deviation, with categorical variables as frequencies. When necessary, logarithmic transformation of the parameters was carried out. The values transformed into logs for analysis were then reconverted for the presentation of the data. 

Student’s *t*-test was used to compare T0 values with T1 values (0 and 12 months). A bidirectional *t*-test for independent samples by ANOVA was used to compare intergroup data. All the analyses were carried out using IBM-SPSS software. The result was considered statistically significant when *p* < 0.05. 

## 3. Results

All assessments were carried out at baseline T0 and at T1. Table 1 lists the DEXA values and biochemical parameters in the control group (CG) and the ospemifene group (OSPG) at T0 and T1. 

In the CG, BMD and T-score decrease was statistically significant in the three bone districts analyzed (FN, TF, and L1–L4); meanwhile, in the OSPG, this significant decrease is present only at the level of FN (*p* = 0.0001 and *p* = 0.0001, for the BMD and T-score, respectively). The BMD and T-score values at TF and L1–l4 levels do not change in the OSPG (*p* = NS and *p* = NS), demonstrating the OSP protective effect on bone loss on TF and L1–L4 (Table 1, upper panel)

A significant increase in 25-OH-D levels was recorded after 12 months of treatment compared to basal levels in the CG (*p* ≤ 0.001) and OSPG (*p* = 0.003), which became optimal (≥30 ng/mL) at 1 year. No significant changes were observed in osteocalcin, ionized calcium, PTH, and phosphorus between T0 and T1 in both groups, while BAP decreased at the end of the study in OSPG (*p* = 0.003) (Table 1, lower panel)

The differences between the absolute and percent delta of the BMD and T score at the three bone levels are shown in Table 2. 

These data confirm the difference in response to the treatment, as OSPG maintains the BMD TF and BMD L1–l4 at baseline levels, while in the CG a worsening was highlighted. Ospemifene, therefore, contributes to the maintenance of BMD-TF and BMD-L1–L4 TF levels, promoting the bone safety profile. Figure 1 summarises one year BMD observation in the control group and in the OSP treated group. 

## 4. Discussion

Bone mineral density and bone architecture are the result of a balance between osteoclastic resorption and osteoblastic formation. Numerous hormonal and dietary factors influence the balance of bone production. Calcium, vitamin D, estrogen, and parathyroid hormone help to maintain bone homeostasis. Osteoclasts resorb bone over a period of weeks and are especially active during periods of rapid remodeling (e.g., after menopause). Because osteoclasts work faster than osteoblasts, the rate of bone loss may outpace the rate of bone production. During menopause transition (MT), the newly produced bone is at increased risk of fracture because it is less densely mineralized, collagen has not matured, and resorption sites are temporarily unfilled. Thus, the lack of sex hormones at menopause contributes significantly to the imbalance between the formation and resorption of the bone matrix [4]. 

The pattern of hormonal changes mirrors those in BMD, with the most rapid increases in FSH and decreases in estradiol occurring in the years around the menopause transition [18].

Therefore, this period of women’s hormonal life—menopause—is critical for changes in bone strength, which sets the stage for the development of osteoporosis and fracture susceptibility in older age [19]. 

Many studies confirm the efficacy of different HRT regimens for the relief of climacteric syndrome and the treatment of osteoporosis [6]. Indeed, HRT can actually be considered as an important tool for the prevention and treatment of both climacteric syndrome and osteoporosis.

However, HRT is not suitable for all postmenopausal women due to its contraindications. Furthermore, the fear of developing breast cancer limits the choice of HRT in many women. The development of new, safe, and alternative therapeutic tools to HRT is an important necessity for the prevention of osteoporosis and the treatment of menopausal symptoms.

Selective estrogen receptor modulators (SERMs) are synthetic non-steroidal agents that have varying estrogen agonist and antagonist activities in different tissues, most likely due to the receptor conformation of structural changes associated with the SERM’s binding and the consequent effect on transcription. Indeed, SERMs are ER ligands that act like estrogens in some tissues, but block estrogen action in others through competitive inhibition of estrogen binding to ERs. 

Ospemifene is a SERM, and a new therapeutic option for the treatment of VVA in postmenopausal women [7] due to its particular stimulating effect on the hormone receptors of the vaginal mucosa, but its neutral effect on the estrogen receptors in the breast and uterus [9].

Clinical studies indicate that OSP has significantly better effects on bone markers than those of placebos, similar to those exerted by oral estrogens and other SERMs (e.g., raloxifene and bazedoxifene) approved for osteoporosis treatment and prevention [20]. 

In experimental studies, the effects of ospemifene on bone were similar to those with raloxifene or estradiol, or in sham-operated versus ovariectomized [OVX] rats. Qu Q e coll. showed that both ospemifene and raloxifene exert bone protective effects, decreasing bone turnover and maintaining bone volume in an OVX rat model [21].

Other studies on rat models confirm the positive effects of ospemifene on bone strength, bone mineral content, BMD, and bone structure [22].

One–two-phase studies on postmenopausal women showed a dose-dependent decrease in bone turnover markers with ospemifene versus a placebo, similar to raloxifene over 12 weeks of treatment [14,15,16].

In a phase 3 study, ospemifene at 60 mg/day for 12 weeks showed improvements in all VVA parameters (first endpoint) and significantly greater decreases in 7 of 9 bone biomarkers versus a placebo (second endpoint) [16].

Lower bone resorption markers in OSP-treated women were observed independently of time since menopause (less or more than five years) or baseline status of bone mineral density (BMD) (normal, osteopenia, or osteoporosis) [23]. 

In this interventional and prospective-control study, ospemifene was administered continuously at a daily dose of 60 mg orally to postmenopausal women (OSPG) with VVA. Bone metabolic biochemical markers and DEXA parameters were compared before (T0) and after 12 months (T1) of treatment, with a matched control group (CG). 

In the CG, a statistically significant decrease in BMD and T-score in the three bone districts analyzed (FN, TF and L1–L4) was found, while in the OSPG only FN (*p* = 0.0001 and *p* = 0.0001) values decrease significantly, while the values of TF and L1–l4 (*p* = NS and *p* = NS) were unchanged, proving OSP’s effectiveness in counteracting bone loss on TF and L1–L4.

As expected, as a consequence of the administration of vitamin D up to sufficient levels in deficient subjects, a significant increase in 25-OH-D levels was recorded after 12 months of treatment compared to basal levels in the CG (*p* ≤ 0 001) and in the OSPG (*p* = 0.003), which are found to be optimal (≥30 ng/mL) at one year. No significant changes were observed in osteocalcin, ionized calcium, PTH, and phosphorus between T0 and T1 in both groups, while BAP decreased at the end of the study in OSP G (*p* = 0.003). Treatment with OSP does not significantly change the values of biochemical markers of bone metabolism. The neutrality of OSP on most biochemical markers can be attributed to the fact that the sample of women examined tends to be healthy. The data relating to BAP confirm the findings of previous studies. This is in agreement with findings reported by other studies, which found an improvement in bone biochemical markers, but in a shorter study period (3 or 6 months) [16,24].

The differences between the absolute and percent delta of BMD and T-score at the three bone levels confirm the variation in response to the treatment, as OSP maintains the BMD TF and BMD L1–L4 at baseline levels, while in the CG, the worsening was highlighted. Thus, ospemifene contributes to maintaining BMD-TF and BMD-L1L4 TF levels, promoting the bone safety profile.

The decrease in BMD values at FN in patients treated with OSP is less than that in the control group. This finding confirms the slowing effect on bone loss at the femoral neck level of OSP treatment compared to the control group. Given the characteristics of the trabecular (histological characteristic) bone of the femoral neck, the slower turnover, and the reduced tropism [25], along with the relative difficulty of reaching this area by drugs, a longer observation period is likely to be necessary to verify the response to OSP treatment. Overall, our data confirm the positive effect of ospemifene on bone metabolism and bone mineral profile in healthy postmenopausal women with VVA. Moreover, our data confirm the safety profile of long-term OSP treatment for the maintenance and improvement of bone mass in healthy postmenopausal women with VVA.

### The Strengths and Weaknesses of the Study

A strength is the fact that our sample represents an unselected population of patients who contacted the clinical center (real-life data). The comparison with a control group with the same characteristics constitutes a further strength of the study. The only fundamental criterion for inclusion was that the patient reported symptoms of VVA.

Since the women enrolled came to our observation for menopausal symptoms, the panel of biochemical markers of bone metabolism is relatively small (real-life data). This may be a weakness of the study. 

## 5. Conclusions

Our data support the hypothesis that long-term treatment with OSP improves bone mineral markers at the total femur and lumbar spine and slows bone loss at the femoral neck level compared to the control group.

Overall, ospemifene therapy was demonstrated to exert a protective and safety effect on bone loss occurring in menopausal transition.

We, therefore, believe that ospemifene can be considered as a safe treatment option for healthy postmenopausal women affected by VVA.

## Figures and Tables

**Figure 1 jcm-11-06316-f001:**
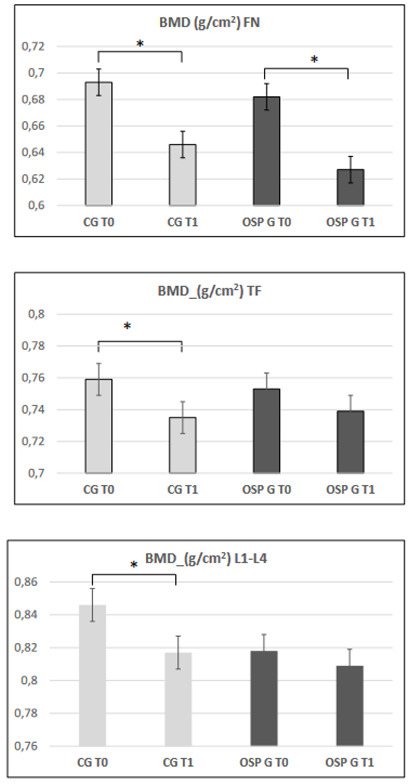
BMD values in FN, TF, and L1–L4 in the two studied groups, namely CG and OSPG. * indicates a statistically significant difference between the variables.

**Table 1 jcm-11-06316-t001:** Bone densitometry assessment and plasma bone turnover biomarkers in the control group (CG) and in the ospemifene group (OSPG) at T0 and T1. Data are expressed as mean ± standard deviation (SD). Abbreviations are as follows: BMD, bone mineral density; L1–L4, lumbar spine; FN, femoral neck; TF, total femur; BMD, bone mineral density; 25-OH-D, serum vitamin D; PTH, parathyroid hormone 1–84; BAP, bone alkaline phosphatase. NS: result not statistically significant.

	CG T0 *n* = 67	CG T1 *n* = 67	*p*	OSPG T0 *n* = 61	OSPG T1 *n* = 61	*P*
**Bone Densitometry Assessment**						
BMD (g/cm^2^) FN	0.693	0.646	0.0001	0.682	0.627	0.0001
T score FN	−1.62	−1.85	0.0001	−1.88	−2.05	0.0001
Z Score FN	−0.35	−0.38	NS	−0.82	−0.91	NS
BMD_(g/cm^2^) TF	0.759	0.735	0.0001	0.753	0.739	NS
T score TF	−1.52	−1.69	0.0001	−1.60	−1.67	NS
Z Score TF	−0.40	−0.43	NS	−0.86	−0.89	NS
BMD (g/cm^2^) L1–L4	0.846	0.817	0.03	0.819	0.809	NS
T score L1–L4	−1.89	−2.10	0.007	−2.31	−2.27	NS
Z Score L1–L4	−0.54	−0.52	NS	−1.26	−1.03	0.001
**Plasma Bone Turnover Biomarkers**						
25-OH-D (ng/mL)	24.94	35.81	0.0001	32.3	38.7	0.003
Osteocalcin (ng/mL)	18.40	19.39	NS	18.85	17.06	NS
Total Calcium (mg/dL)	9.31	9.44	0.044	9.32	9.37	NS
Ionized Calcium	1.23	1.24	NS	1.24	1.23	NS
PTH (pg/mL)	27.62	26.05	NS	25.77	26.12	NS
BAP (g/L)	12.93	12.25	NS	13.93	10.21	0.003
Phosphorus (mg/dL)	3.45	3.40	NS	3.41	3.44	NS

**Table 2 jcm-11-06316-t002:** Differences between absolute and percent delta of BMD and T-score at FN, TF, and L1–L4. *p* in italics indicates a tendency towards statistical significance.

	Delta CG	Delta OSPG	*p*	Delta % CG	Delta % OSPG	*p*
BMD (g/cm^2^) FN	−0.047	−0.055	NS	6.0%	7.3%	NS
T score FN	−0.23	−0.17	NS	23.8%	15.0%	NS
BMD (g/cm^2^) TF	−0.024	−0.005	0.033	2.9%	0.4%	0.0294
T score TF	−0.17	−0.07	NS	20.4%	10.2%	NS
BMD (g/cm^2^) L1–L4	−0.029	−0.010	0.045	3.1%	0.3%	*0.067*
T score L1–L4	−0.21	0.04	0.01	11.0%	3.3%	NS

## Data Availability

Data supporting reported results can be found at Italian National Research Council Gabriele Monasterio Tuscan Foundation in Pisa, Italy.

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
