# Peer review of "Effect of Ospemifene on Densitometric and Plasma Bone Metabolism Biomarkers in Postmenopausal Women Reporting Vulvar and Vaginal Atrophy (VVA)"

_jcm, 2022, doi:10.3390/jcm11216316_

Round 1
Reviewer 1 Report
The manuscript presents interesting research but needs to be revised
1. Please provide exact definitions of T-score and Z-score in Study design.
2. L105 - L106 - it should be 2.5, not 2 5
3. What is '' q '' in Table 2 in column ‘’Delta OSP G’’ .
4. I suggest verify the data in Table 2. The delta CG data for TF BMD and delta OSP G for T score L1-L4 are correct? The reviewer did not calculate the delta%.
5. Figure 1 is not described. Moreover, in my opinion Figure 1 is unnecessary as the same results are shown in Table 1
6. L170 – The oestrogen is a sex hormone.
7. L183-L186 is a repeat of L54-L56
8. L233-L235 - Is the item Archer et al. [22] is suitable? This part needs to be corrected
9. Conclusion – L261-L263 - to remove
Author Response
Abstract/ Introduction
In meta data bone min-eral density (BMD)-- should be Mineral
HRT term used-- Need to use Full name 1st time
Thanks for your comment: we have made changes to the text.
Material & Method :
Study period means the year not mentioned. as if the study conducted during or after COVID-- the Vit D dose have different dose.
The study was conducted before the COVID therefore the covid does not represent a bias in the values of vitamin D described.
Result
Table 1 and Table 2 is there with description.
Please refer Table 1 and Table 2 to describe -- so that easy to understand what the table states.
Discussion
Thank you for your suggestion. We have made the relative changes in the text (line 143-153)
Repeatation of
However, not all menopausal women are eligible for HRT, due to contraindications to hormone therapy. Some women refuse HRT for fear of breast cancer. There is therefore a need for new therapeutic options to prevent osteoporosis and treat menopausal symptoms. with introduction section.
Thanks for the comment. we have reformulated the sentence in order to reiterate the concept already expressed in the introduction. (line 195-199)

Reviewer 2 Report
Abstract:
In meta data bone min-eral density (BMD)-- should be Mineral
Introduction
HRT term used-- Need to use Full name 1st time
Material & Method :
Study period means the year not mentioned.
as if the study conducted during or after COVID-- the Vit D dose have different dose.
Result
Table 1 and Table 2 is there with description.
Please refer Table 1 and Table 2 to describe -- so that easy to understand what the table states.
Discussion
Repeatation of
However, not all menopausal women are eligible for HRT, due to contraindications to hormone therapy. Some women refuse HRT for fear of breast cancer. There is therefore a need for new therapeutic options to prevent osteoporosis and treat menopausal symptoms. with introduction section.
Author stated, Other studies on animal models-- which animal need to specify.
Side effect of the treatment has not been illustrated. Authors are requested to highlight the side effects if they have noticed during study.
References:
There is no 2020 reference
2022 only one
Examples can be added:
Pingarrón Santofímia, C., Lafuente González, P., Guitiérrez Vélez, M. D. C., Calvente Aguilar, V., Poyo Torcal, S., Terol Sánchez, P., & Palacios, S. (2022). Long-term use of ospemifene in clinical practice for vulvo-vaginal atrophy: end results at 12 months of follow-up. Gynecological Endocrinology, 1-6.
Pingarrón, C., Lafuente, P., Poyo Torcal, S., Lopez Verdu, H., Martinez Garcia, M. S., & Palacios, S. (2022). Vaginal health, endometrial thickness and changes in bone markers in postmenopausal women after 6 months of treatment with ospemifene in real clinical practice. Gynecological Endocrinology, 38(1), 78-82.
Pup, L. D., & Sánchez-Borrego, R. (2020). Ospemifene efficacy and safety data in women with vulvovaginal atrophy. Gynecological Endocrinology, 36(7), 569-577.
Requested to add more recent references and
Take a relation with COVID - Vit D therapy reference in discussion section.
Author Response
The manuscript presents interesting research but needs to be revised
- Please provide exact definitions of T-score and Z-score in Study design.
Thanks for the comment, we added the definitions and reference in the text (line 103-110)
- L105 - L106 - it should be 2.5, not 2 5. Thanks for the comment, we changed at 2.5
- What is '' q '' in Table 2 in column ‘’Delta OSP G’’.4. I suggest verify the data in Table 2. The delta CG data for TF BMD and delta OSP G for T score L1-L4 are correct? The reviewer did not calculate the delta%.
Thanks for the comments num3 and 4. We changed the incorrect data in the table 2.
- Figure 1 is not described. Moreover, in my opinion Figure 1 is unnecessary as the same results are shown in Table 1
Thanks for the appropriate observation. We have returned to change the text by separating the table from the figure that shows the summary data on the BMD more clearly.
- L170 – The oestrogen is a sex hormone.
Thanks: we have changed the text removing “The oestrogen”
- L183-L186 is a repeat of L54-L56
Thanks: we changed the sentence.
- L233-L235 - Is the item Archer et al. [22] is suitable? This part needs to be corrected
Thanks: we changed the reference.
- Conclusion – L261-L263 - to remove
Thanks: Done, we removed the sentence “The present study evaluated the effects of Ospemifene at a dose of 60 mg per day orally on bone mineral values and on bone metabolism in healthy postmenopausal women affected by VVA”.
Author stated, Other studies on animal models-- which animal need to specify.
thanks for your clarification request: we have corrected the text (258-264).
Side effect of the treatment has not been illustrated. Authors are requested to highlight the side effects if they have noticed during study.
thanks for this consideration. We did not report side effects in the text because they were not observed during the study
References:
There is no 2020 reference:
We considered to add reference 25:
Pingarrón Santofímia, C., Lafuente González, P., Guitiérrez Vélez, M. D. C., Calvente Aguilar, V., Poyo Torcal, S., Terol Sánchez, P., & Palacios, S. (2022). Long-term use of ospemifene in clinical practice for vulvo-vaginal atrophy: end results at 12 months of follow-up. Gynecological Endocrinology, 1-6.
